# Explanations can be manipulated and geometry is to blame

**Ann-Kathrin Dombrowski**[1]**, Maximilian Alber**[5]**, Christopher J. Anders**[1]**,**
**Marcel Ackermann**[2]**, Klaus-Robert Müller**[1,3,4]**, Pan Kessel**[1]

[1]Machine Learning Group, Technische Universität Berlin, Germany
[2]Department of Video Coding & Analytics, Fraunhofer Heinrich-Hertz-Institute, Berlin, Germany
[3]Max-Planck-Institut für Informatik, Saarbrücken, Germany
[4]Department of Brain and Cognitive Engineering, Korea University, Seoul, Korea
[5]Charité Berlin, Berlin, Germany

## Abstract

Explanation methods aim to make neural networks more trustworthy and interpretable. In this paper, we demonstrate a property of explanation methods which is disconcerting for both of these purposes. Namely, we show that explanations can be manipulated *arbitrarily* by applying visually hardly perceptible perturbations to the input that keep the network's output approximately constant. We establish theoretically that this phenomenon can be related to certain geometrical properties of neural networks. This allows us to derive an upper bound on the susceptibility of explanations to manipulations. Based on this result, we propose effective mechanisms to enhance the robustness of explanations.

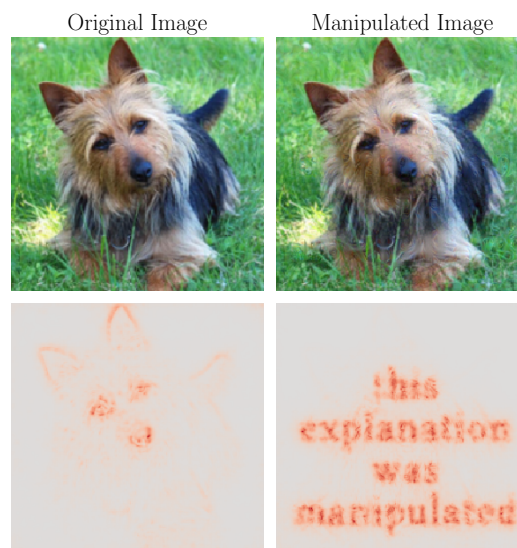

Figure 1: Original image with corresponding explanation map on the left. Manipulated image with its explanation on the right. The chosen target explanation was an image with a text stating "this explanation was manipulated".

# 1 Introduction

Explanation methods have attracted significant attention over the last years due to their promise to open the black box of deep neural networks. Interpretability is crucial for scientific understanding and safety critical applications.

Explanations can be provided in terms of explanation maps[1–20] that visualize the relevance attributed to each input feature for the overall classification result. In this work, we establish that these explanation maps can be changed to an *arbitrary target map*. This is done by applying a visually hardly perceptible perturbation to the input. We refer to Figure 1 for an example. This perturbation does not change the output of the neural network, i.e. in addition to the classification result also the vector of all class probabilities is (approximately) the same.

This finding is clearly problematic if a user, say a medical doctor, is expecting a robustly interpretable explanation map to rely on in the clinical decision making process.

Motivated by this unexpected observation, we provide a theoretical analysis that establishes a relation of this phenomenon to the geometry of the neural network's output manifold. This novel understanding allows us to derive a bound on the degree of possible manipulation of the explanation map. This bound is proportional to two differential geometric quantities: the principle curvatures and the geodesic distance between the original input and its manipulated counterpart. Given this theoretical insight, we propose efficient ways to limit possible manipulations and thus enhance resilience of explanation methods.

In summary, this work provides the following key contributions:

- We propose an algorithm which allows to manipulate an image with a hardly perceptible perturbation such that the explanation matches an arbitrary target map. We demonstrate its effectiveness for six different explanation methods and on four network architectures as well as two datasets.
- We provide a theoretical understanding of this phenomenon for gradient-based methods in terms of differential geometry. We derive a bound on the principle curvatures of the hypersurface of equal network output. This implies a constraint on the maximal change of the explanation map due to small perturbations.
- Using these insights, we propose methods to undo the manipulations and increase the robustness of explanation maps by smoothing the explanation method. We demonstrate experimentally that smoothing leads to increased robustness not only for gradient but also for propagation-based methods.

## 1.1 Related work

In [21], it was demonstrated that explanation maps can be sensitive to small perturbations in the image. The authors apply perturbations to the image which lead to an unstructured change in the explanation map. Specifically, their approach can increase the overall sum of relevances in a certain region of the explanation map. Our work focuses on structured manipulations instead, i.e. to reproduce a given target map on a pixel-by-pixel basis. Furthermore, their attacks only keep the classification result the same which often leads to significant changes in the network output. From their analysis, it is therefore not clear whether the explanation *or* the network is vulnerable (and the explanation map simply reflects the relevance of the perturbation faithfully). Our method keeps the output of the network (approximately) constant. We furthermore provide a theoretical analysis in terms of differential geometry and propose effective defense mechanisms. Another approach [22] adds a constant shift to the input image, which is then eliminated by changing the bias of the first layer. For some methods, this leads to a change in the explanation map. Contrary to our approach, this requires to change the network's biases. In [23], explanation maps are changed by randomization of (some of) the network weights and in [24] the complete network is fine-tuned to produce manipulated explanations while the accuracy remains high. These two approaches are different from our method as they do not aim to change the explanation to a specific target explanation map and modify the parameters of the network. In [25, 26], it is proposed to bound the (local) Lipschitz constant of the explanation. This has the disadvantage that explanations become insensitive to *any* small perturbation, e.g. even those which lead to a substantial change in network output. This is clearly undesirable as the explanation should reflect why the perturbation leads to such a drastic change of the network's

confidence. In this work, we therefore propose to only bound the curvature of the hypersurface of equal network output.

## 2 Manipulation of explanations

### 2.1 Explanation methods

We consider a neural network $g : \mathbb{R}^d \to \mathbb{R}^K$ with relu non-linearities which classifies an image $x \in \mathbb{R}^d$ in $K$ categories with the predicted class given by $k = \arg\max_i g(x)_i$. The explanation map is denoted by $h : \mathbb{R}^d \to \mathbb{R}^d$ and associates an image with a vector of the same dimension whose components encode the relevance score of each pixel for the neural network's prediction.

Throughout this paper, we will use the following explanation methods:

- **Gradient**: The map $h(x) = \frac{\partial g}{\partial x}(x)$ is used and quantifies how infinitesimal perturbations in each pixel change the prediction $g(x)$ [2, 1].

- **Gradient × Input**: This method uses the map $h(x) = x \odot \frac{\partial g}{\partial x}(x)$ [14]. For linear models, this measure gives the exact contribution of each pixel to the prediction.

- **Integrated Gradients**: This method defines $h(x) = (x - \bar{x}) \odot \int_0^1 \frac{\partial g(\bar{x}+t(x-\bar{x}))}{\partial x} \mathrm{d}t$ where $\bar{x}$ is a suitable baseline. See the original reference [13] for more details.

- **Guided Backpropagation (GBP)**: This method is a variation of the gradient explanation for which negative components of the gradient are set to zero while backpropagating through the non-linearities [4].

- **Layer-wise Relevance Propagation (LRP)**: This method [5, 16] propagates relevance backwards through the network. For the output layer, relevance is defined by[1]

$$R_i^L = \delta_{i,k} \, , \tag{1}$$

which is then propagated backwards through all layers but the first using the $z^+$ rule

$$R_i^l = \sum_j \frac{x_i^l (W^l)_{ji}^+}{\sum_i x_i^l (W^l)_{ji}^+} R_j^{l+1} \, , \tag{2}$$

where $(W^l)^+$ denotes the positive weights of the $l$-th layer and $x^l$ is the activation vector of the $l$-th layer. For the first layer, we use the $z^{\mathcal{B}}$ rule to account for the bounded input domain

$$R_i^0 = \sum_j \frac{x_j^0 W_{ji}^0 - l_j (W^0)_{ji}^+ - h_j (W^0)_{ji}^-}{\sum_i (x_j^0 W_{ji}^0 - l_j (W^0)_{ji}^+ - h_j (W^0)_{ji}^-)} R_j^1 \, , \tag{3}$$

where $l_i$ and $h_i$ are the lower and upper bounds of the input domain respectively.

- **Pattern Attribution (PA)**: This method is equivalent to standard backpropagation upon element-wise multiplication of the weights $W^l$ with learned patterns $A^l$. We refer to the original publication for more details [17].

These methods cover two classes of attribution methods, namely *gradient-based* and *propagation-based* explanations, and are frequently used in practice [27, 28].

### 2.2 Manipulation Method

For a given explanation method and specified target $h^t \in \mathbb{R}^d$, a manipulated image $x_{\mathrm{adv}} = x + \delta x$ has the following properties:

1. The output of the network stays approximately constant, i.e. $g(x_{\mathrm{adv}}) \approx g(x)$.

2. The explanation is close to the target map, i.e. $h(x_{\mathrm{adv}}) \approx h^t$.

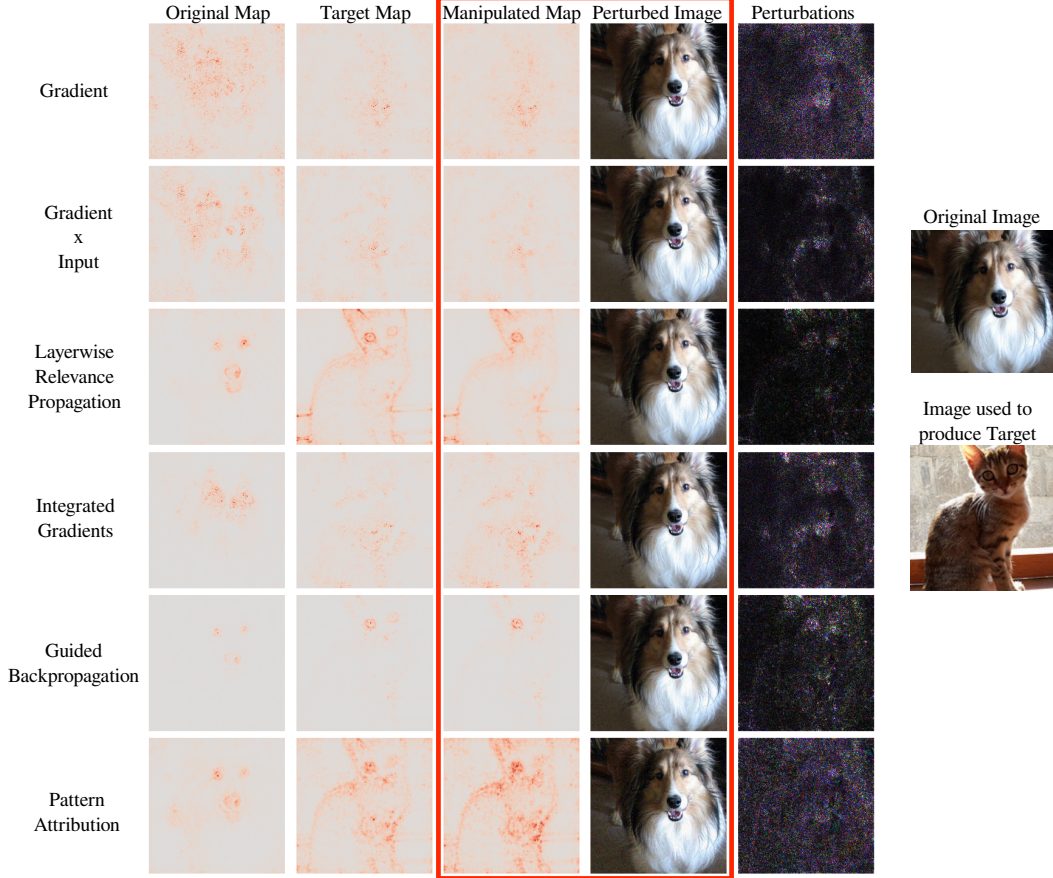

Figure 2: The explanation map of the cat is used as the target and the image of the dog is perturbed. The red box contains the manipulated images and the corresponding explanations. The first column corresponds to the original explanations of the unperturbed dog image. The target map, shown in the second column, is the corresponding explanation of the cat image. The last column visualizes the perturbations.

    3. The norm of the perturbation $\delta x$ added to the input image is small, i.e. $\|\delta x\| = \|x_{\text{adv}} - x\| \ll 1$ and therefore not perceptible.

We obtain such manipulations by optimizing the loss function

$$\mathcal{L} = \left\| h(x_{\text{adv}}) - h^t \right\|^2 + \gamma \left\| g(x_{\text{adv}}) - g(x) \right\|^2 , \tag{4}$$

with respect to $x_{\text{adv}}$ using gradient descent. We clamp $x_{\text{adv}}$ after each iteration so that it is a valid image. The first term in the loss function (4) ensures that the manipulated explanation map is close to the target while the second term encourages the network to have the same output. The relative weighting of these two summands is controlled by the hyperparameter $\gamma \in \mathbb{R}_+$.

Our method therefore requires us to calculate the gradient with respect to the input $\nabla h(x)$ of the explanation. For relu-networks, this gradient often depends on the vanishing second derivative of non-linearities which leads to problems during optimization of the loss (4). As an example, the gradient method leads to

$$\partial_{x_{\text{adv}}} \left\| h(x_{\text{adv}}) - h^t \right\|^2 \propto \frac{\partial h}{\partial x_{\text{adv}}} = \frac{\partial^2 g}{\partial x_{\text{adv}}^2} \propto \text{relu}'' = 0 \, .$$

We therefore replace the relu with softplus non-linearities

$$\text{softplus}_\beta(x) = \frac{1}{\beta} \log(1 + e^{\beta x}) \, . \tag{5}$$

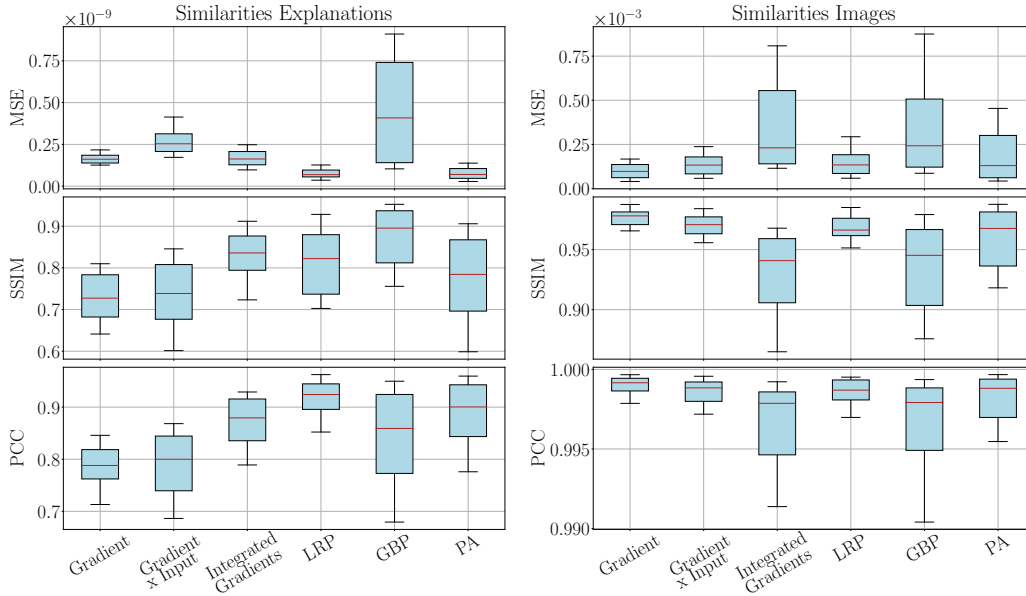

Figure 3: Left: Similarity measures between target $h^t$ and manipulated explanation map $h(x_{adv})$. Right: Similarity measures between original $x$ and perturbed image $x_{adv}$. For SSIM and PCC large values indicate high similarity while for MSE small values correspond to similar images. For fair comparison, we use the same 100 randomly selected images for each explanation method.

For large $\beta$ values, the softplus approximates the relu closely but has a well-defined second derivative. After optimization is complete, we test the manipulated image with the original relu network.

**Similarity metrics:** In our analysis, we assess the similarity between both images and explanation maps. To this end, we use three metrics following [23]: the structural similarity index (SSIM), the Pearson correlation coefficient (PCC) and the mean squared error (MSE). SSIM and PCC are relative similarity measures with values in $[0, 1]$, where larger values indicate high similarity. The MSE is an absolute error measure for which values close to zero indicate high similarity. We normalize the sum of the explanation maps to be one and the images to have values between 0 and 1.

### 2.3 Experiments

To evaluate our approach, we apply our algorithm to 100 randomly selected images for each explanation method. We use a pre-trained VGG-16 network [29] and the ImageNet dataset [30]. For each run, we randomly select two images from the test set. One of the two images is used to generate a target explanation map $h^t$. The other image is perturbed by our algorithm with the goal of replicating the target $h^t$ using a few hundred iterations of gradient descent. We sum over the absolute values of the channels of the explanation map to get the relevance per pixel. Further details about the experiments are summarized in Supplement A.

**Qualitative analysis:** Our method is illustrated in Figure 2 in which a dog image is manipulated in order to have an explanation resembling a cat. For all explanation methods, the target is closely emulated and the perturbation of the dog image is small. More examples can be found in the supplement.

**Quantitative analysis:** Figure 3 shows similarity measures between the target $h^t$ and the manipulated explanation map $h(x_{adv})$ as well as between the original image $x$ and perturbed image $x_{adv}$.[2] All considered metrics show that the perturbed images have an explanation closely resembling the targets. At the same time, the perturbed images are very similar to the corresponding original images. We also verified by visual inspection that the results look very similar. We have uploaded the results of all

runs so that interested readers can assess their similarity themselves[3] and provide code[4] to reproduce them. In addition, the output of the neural network is approximately unchanged by the perturbations, i.e. the classification of all examples is unchanged and the median of $\|g(x_{\text{adv}}) - g(x)\|$ is of the order of magnitude $10^{-3}$ for all methods. See Supplement B for further details.

**Other architectures and datasets:** We checked that comparable results are obtained for ResNet-18 [31], AlexNet [32] and Densenet-121 [33]. Moreover, we also successfully tested our algorithm on the CIFAR-10 dataset [34]. We refer to the Supplement C for further details.

## 3 Theoretical considerations

In this section, we analyze the vulnerability of explanations theoretically. We argue that this phenomenon can be related to the large curvature of the output manifold of the neural network. We focus on the gradient method starting with an intuitive discussion before developing mathematically precise statements.

We have demonstrated that one can drastically change the explanation map while keeping the output of the neural network constant

$$g(x + \delta x) = g(x) = c \tag{6}$$

using only a small perturbation in the input $\delta x$. The perturbed image $x_{\text{adv}} = x + \delta x$ therefore lies on the hypersurface of constant network output $S = \{p \in \mathbb{R}^d | g(p) = c\}$.[5] We can exclusively consider the winning class output, i.e. $g(x) := g(x)_k$ with $k = \arg\max_i g(x)_i$ because the gradient method only depends on this component of the output. Therefore, the hypersurface $S$ is of co-dimension one. The gradient $\nabla g$ for every $p \in S$ is normal to this hypersurface. The fact that the normal vector $\nabla g$ can be drastically changed by slightly perturbing the input along the hypersurface $S$ suggests that the curvature of $S$ is large.

While the latter statement may seem intuitive, it requires non-trivial concepts of differential geometry to make it precise, in particular the notion of the second fundamental form. We will briefly summarize these concepts in the following (see e.g. [35] for a standard textbook). To this end, it is advantageous to consider a normalized version of the gradient method

$$n(x) = \frac{\nabla g(x)}{\|\nabla g(x)\|}. \tag{7}$$

This normalization is merely conventional as it does not change the relative importance of any pixel with respect to the others. For any point $p \in S$, we define the tangent space $T_p S$ as the vector space spanned by the tangent vectors $\dot{\gamma}(0) = \frac{d}{dt}\gamma(t)|_{t=0}$ of all possible curves $\gamma : \mathbb{R} \to S$ with $\gamma(0) = p$. For $u, v \in T_p S$, we denote their inner product by $\langle u, v \rangle$. For any $u \in T_p S$, the *directional derivative* of a function $f$ is uniquely defined for any choice of $\gamma$ by

$$D_u f(p) = \left. \frac{d}{dt} f(\gamma(t)) \right|_{t=0} \qquad \text{with} \qquad \gamma(0) = p \ \text{and} \ \dot{\gamma}(0) = u. \tag{8}$$

We then define the *Weingarten map* as[6]

$$L : \begin{cases} T_p S & \to T_p S \\ u & \mapsto -D_u n(p), \end{cases}$$

where the unit normal $n(p)$ can be written as (7). This map quantifies how much the unit normal changes as we infinitesimally move away from $p$ in the direction $u$. The *second fundamental form* is then given by

$$\mathcal{L} : \begin{cases} T_p S \times T_p S & \to \mathbb{R} \\ u, v & \mapsto -\langle v, L(u) \rangle = \langle v, D_u n(p) \rangle. \end{cases}$$

It can be shown that the second fundamental form is bilinear and symmetric $\mathcal{L}(u,v) = \mathcal{L}(v,u)$. It is therefore diagonalizable with real eigenvalues $\lambda_1, \dots \lambda_{d-1}$ which are called *principle curvatures*.

We have therefore established the remarkable fact that the sensitivity of the gradient map (7) is described by the principle curvatures, a key concept of differential geometry.

In particular, this allows us to derive an upper bound on the maximal change of the gradient map $h(x) = n(x)$ as we move slightly on $S$. To this end, we define the *geodesic distance* $d_g(p,q)$ of two points $p, q \in S$ as the length of the shortest curve on $S$ connecting $p$ and $q$. In the supplement, we show that:

**Theorem 1** *Let $g : \mathbb{R}^d \to \mathbb{R}$ be a network with softplus$_\beta$ non-linearities and $\mathcal{U}_\epsilon(p) = \{x \in \mathbb{R}^d; \|x - p\| < \epsilon\}$ an environment of a point $p \in S$ such that $\mathcal{U}_\epsilon(p) \cap S$ is fully connected. Let g have bounded derivatives $\|\nabla g(x)\| \geq c$ for all $x \in \mathcal{U}_\epsilon(p) \cap S$. It then follows for all $p_0 \in \mathcal{U}_\epsilon(p) \cap S$ that*

$$\|h(p) - h(p_0)\| \leq |\lambda_{max}| \, d_g(p, p_0) \leq \beta \, C \, d_g(p, p_0), \tag{9}$$

*where $\lambda_{max}$ is the principle curvature with the largest absolute value for any point in $\mathcal{U}_\epsilon(p) \cap S$ and the constant $C > 0$ depends on the weights of the neural network.*

This theorem can intuitively be motivated as follows: for relu non-linearities, the lines of equal network output are piece-wise linear and therefore have kinks, i.e. points of divergent curvature. These relu non-linearities are well approximated by softplus non-linearities (5) with large $\beta$. Reducing $\beta$ smoothes out the kinks and therefore leads to reduced maximal curvature, i.e. $|\lambda_{max}| \leq \beta \, C$. For each point on the geodesic curve connecting $p$ and $p_0$, the normal can at worst be affected by the maximal curvature, i.e. the change in explanation is bounded by $|\lambda_{max}| \, d_g(p, p_0)$.

There are two important lessons to be learnt from this theorem: the geodesic distance can be substantially greater than the Euclidean distance for curved manifolds. In this case, inputs which are very similar to each other, i.e. the Euclidean distance is small, can have explanations that are drastically different. Secondly, the upper bound is proportional to the $\beta$ parameter of the softplus non-linearity. Therefore, smaller values of $\beta$ provably result in increased robustness with respect to manipulations.

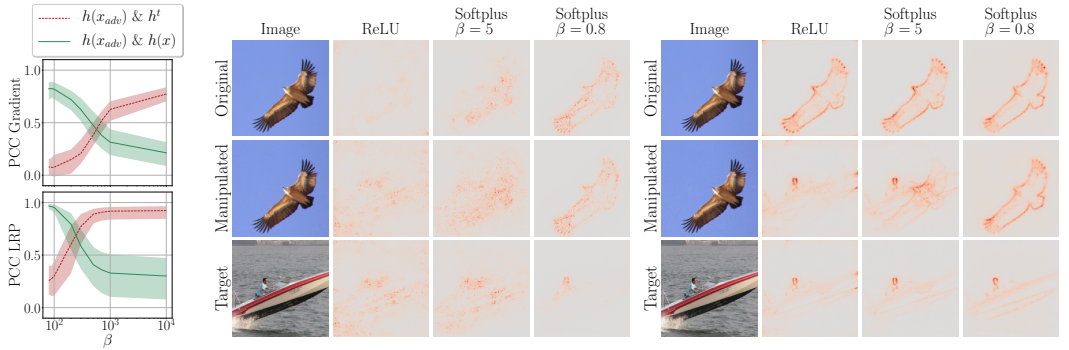

Figure 4: Left: $\beta$ dependence for the correlations of the manipulated explanation (here Gradient and LRP) with the target and original explanation. Lines denote the medians, $10^{th}$ and $90^{th}$ percentiles are shown in semitransparent colour. Center and Right: network input and the respective explanation maps as $\beta$ is decreased for Gradient (center) and LRP (right).

## 4 Robust explanations

Using the fact that the upper bound of the last section is proportional to the $\beta$ parameter of the softplus non-linearities, we propose $\beta$-*smoothing* of explanations. This method calculates an explanation using a network for which the relu non-linearities are replaced by softplus with a small $\beta$ parameter to smooth the principle curvatures. The precise value of $\beta$ is a hyperparameter of the method, but we find that a value around one works well in practice.

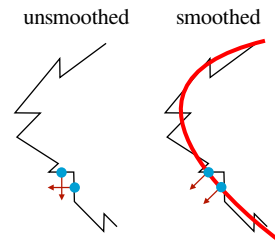

As shown in the supplement, a relation between SmoothGrad [12] and $\beta$-smoothing can be proven for a one-layer neural network:

**Theorem 2** *For a one-layer neural network $g(x) = relu(w^T x)$ and its $\beta$-smoothed counterpart $g_\beta(x) = softplus_\beta(w^T x)$, it holds that*

$$\mathbb{E}_{\epsilon \sim p_\beta} \left[ \nabla g(x - \epsilon) \right] = \nabla g_{\frac{\beta}{\|w\|}}(x),$$

*where $p_\beta(\epsilon) = \frac{\beta}{(e^{\beta\epsilon/2} + e^{-\beta\epsilon/2})^2}$.*

Since $p_\beta(x)$ closely resembles a normal distribution with variance $\sigma = \log(2)\frac{\sqrt{2\pi}}{\beta}$, $\beta$-smoothing can be understood as $N \to \infty$ limit of SmoothGrad $h(x) = \frac{1}{N}\sum_{i=1}^{N} \nabla g(x - \epsilon_i)$ where $\epsilon_i \sim g_\beta \approx \mathcal{N}(0, \sigma)$. We emphasize that the theorem only holds for a one-layer neural network, but for deeper networks we empirically observe that both lead to visually similar maps as they are considerably less noisy than the gradient map. The theorem therefore suggests that SmoothGrad can similarly be used to smooth the curvatures and can thereby make explanations more robust.[7]

**Experiments:** Figure 4 demonstrates that $\beta$-smoothing allows us to recover the original explanation map by decreasing the value of the $\beta$ parameter. We stress that this works for all considered methods. We also note that the same effect can be observed using SmoothGrad by successively increasing the standard deviation $\sigma$ of the noise distribution. This further underlines the similarity between the two smoothing methods.

If an attacker knew that smoothing was used to undo the manipulation, they could try to attack the smoothed method directly. However, both $\beta$-smoothing and SmoothGrad are substantially more robust than their non-smoothed counterparts, see Figure 5. It is important to note that $\beta$-smoothing achieves this at considerably lower computational cost: $\beta$-smoothing only requires a single forward and backward pass, while SmoothGrad requires as many as the number of noise samples (typically between 10 to 50).

We refer to Supplement D for more details on these experiments.

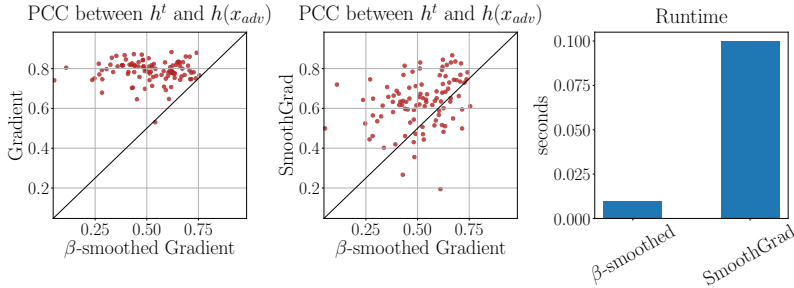

Figure 5: Left: markers are clearly left of the diagonal, i.e. explanations are more robust to manipulations when $\beta$-smoothing is used. Center: SmoothGrad has comparable results to $\beta$-smoothing, i.e. markers are distributed around the diagonal. Right: $\beta$-smoothing has significantly lower computational cost than SmoothGrad.

Figure 6 shows the evolution of the gradient explanation maps when reducing the $\beta$ parameter of the softplus activations. We note that for small $\beta$ the explanation maps tend to become similar to LRP/GBP/PA explanation maps (see Figure 2 for comparison). Figure 7 demonstrates that $\beta$-smoothing leads to better performance than the gradient method and to comparable performance with SmoothGrad on the pixel-flipping metric [5, 36].

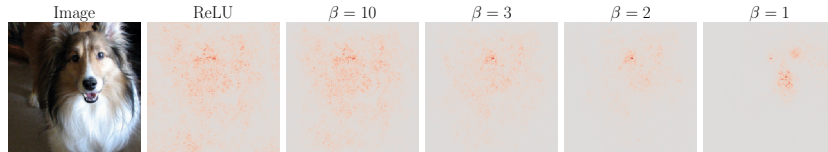

Figure 6: Gradient explanation map produced with the original network and a network with softplus activation functions using various values for $\beta$.

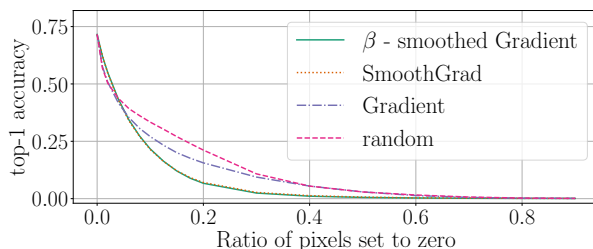

Figure 7: Pixelflipping performance compared to random baseline (the lower the accuracy the better the explanation): the metric sorts pixels of images by relevance and incrementally sets the pixels to zero starting with the most relevant. In each step, the network's performance is evaluated on the complete ImageNet validation set.

## 5  Conclusion

Explanation methods have recently become increasingly popular among practitioners. In this contribution, we show that dedicated imperceptible manipulations of the input data can yield arbitrary and drastic changes of the explanation map. We demonstrate both qualitatively and quantitatively that explanation maps of many popular explanation methods can be arbitrarily manipulated. Crucially, this can be achieved while keeping the model's output constant. A novel theoretical analysis reveals that in fact the large curvature of the network's decision function is one important culprit for this unexpected vulnerability. Using this theoretical insight, we can profoundly increase the resilience to manipulations by smoothing *only* the explanation process while leaving the model itself unchanged.

Future work will investigate possibilities to modify the training process of neural networks itself such that they can become less vulnerable to manipulations of explanations. Another interesting future direction is to generalize our theoretical analysis of gradient-based to propagation-based methods. This seems particularly promising because our experiments strongly suggest that similar theoretical findings should also hold for these explanation methods.

**Acknowledgments**

We want to thank the anonymous reviewers for their helpful feedback. We also thank Kristof Schütt, Grégoire Montavon and Shinichi Nakajima for useful discussions. This work is supported by the German Ministry for Education and Research as Berlin Big Data Center (01IS18025A) and Berlin Center for Machine Learning (01IS18037I). This work is also supported by the Information & Communications Technology Planning & Evaluation (IITP) grant funded by the Korea government (No. 2017-0-001779), as well as by the Research Training Group "Differential Equation- and Data-driven Models in Life Sciences and Fluid Dynamics (DAEDALUS)" (GRK 2433) and Grant Math+, EXC 2046/1, Project ID 390685689 both funded by the German Research Foundation (DFG).

## Footnotes

[1]Here we use the Kronecker symbol $\delta_{i,k} = \begin{cases} 1, & \text{for } i = k \\ 0, & \text{for } i \neq k \end{cases}$.

[2]Throughout this paper, boxes denote 25th and 75th percentiles, whiskers denote 10th and 90th percentiles, and solid lines show the medians

[3]https://drive.google.com/drive/folders/1TZeWngoevHRuIw6gb5CZDIRrc7EWf5yb?usp=sharing

[4]https://github.com/pankessel/adv_explanation_ref

[5]It is sufficient to consider the hypersurface $S$ in a neighbourhood of the unperturbed input $x$.

[6]The fact that $D_u n(p) \in T_p S$ follows by taking the directional derivative with respect to $u$ on both sides of $\langle n, n \rangle = 1$.

[7]For explanation methods $h(x)$ other than gradient, SmoothGrad needs to be used in a slightly generalized form, i.e. $\frac{1}{N}\sum_{i=1}^{N} h(x - \epsilon_i)$.

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
