[Supplementary Material]

# Supplement

## Contents

## A  Details on experiments

We provide a *run_attack.py* file in our reference implementation which allows one to produce manipulated images. The hyperparameter choices used in our quantitative analysis are summarized in Table 1. We set $\beta_0 = 10$ and $\beta_e = 100$ for beta growth (see section below for a description). The column 'factors' summarizes the weighting of the mean squared error of the explanation maps and the network outputs respectively.

| method | iterations | lr | factors |
|--------|-----------|-----|---------|
| Gradient | 1500 | $10^{-3}$ | $10^{11}$, $10^6$ |
| Grad x Input | 1500 | $10^{-3}$ | $10^{11}$, $10^6$ |
| IntGrad | 500 | $5 \times 10^{-3}$ | $10^{11}$, $10^6$ |
| LRP | 1500 | $2 \times 10^{-4}$ | $10^{11}$, $10^6$ |
| GBP | 1500 | $10^{-3}$ | $10^{11}$, $10^6$ |
| PA | 1500 | $2 \times 10^{-3}$ | $10^{11}$, $10^6$ |

Table 1: Hyperparameters used in our analysis.

The patterns for explanation method PA are trained on a subset of the ImageNet training set. The baseline $\bar{x}$ for explanation method IG was set to zero. To approximate the integral, we use 30 steps for which we verified that the attributions approximately adds up to the score at the input.

## A.1 Beta growth

In practise, we observe that we get slightly better results by increasing the value of $\beta$ of the softplus $sp(x) = \frac{1}{\beta} \ln\left(1 + e^{\beta x}\right)$ during training a start value $\beta_0$ to a final value $\beta_e$ using

$$\beta(t) = \beta_0 \left(\frac{\beta_e}{\beta_0}\right)^{t/T} , \tag{1}$$

where $t$ is the current optimization step and $T$ denotes the total number of steps. Figure 1 shows the MSE for images and explanation maps during training with and without $\beta$-growth. This strategy is however not essential for our results.

Figure 1: MSE between $x$ and $x_{adv}$ (left) and between $h^t$ and $h(x_{adv})$ (right) for various values for $\beta$.

We use beta growth for all methods except LRP for which we do not find any speed-up in the optimization as the LRP rules do not explicitly depend on the second derivative of the relu activations. Figure 2 demonstrates that for large beta values the softplus networks approximate the relu network well. Figure 3 and Figure 4 show this for an example for the gradient and the LRP explanation method. We also note that for small beta the gradient explanation maps become more similar to LRP/GPB/PA explanation maps.

Figure 2: Error measures between the gradient explanation map produced with the original network and explanation maps produced with a network with softplus activation functions using various values for $\beta$.

Figure 3: Gradient explanation map produced with the original network and a network with softplus activation functions using various values for $\beta$.

Figure 4: LRP explanation map produced with the original network and a network with softplus activation functions using various values for $\beta$.

## A.2 Comparison with similar work

Ghorbani et al. (2017) present a targeted but unstructured attack to manipulate explanations. They perturb the original image by a fixed stepsize in the direction of the signed gradient $\mathrm{sgn}(\nabla_x D)$ of the dissimilarity function

$$D(\boldsymbol{x}) = \sum_{i \in \mathcal{A}} \boldsymbol{I}(\boldsymbol{x_i}). \tag{2}$$

This increases the accumulated relevance in the specified area $\mathcal{A}$ but cannot structurally reproduce a target explanation except for very selected and simple cases. We demonstrate this qualitatively in Figure 5 (a-c) and quantitatively for 100 test images by comparing the Pearson Correlation Coefficient between the explanations prior and post the manipulation on the left hand side of Figure 5 (d). Furthermore our method keeps the output constant while their method often leads to a significant change in confidence, see right hand side of Figure 5 (d). Constant output is crucial for the geometrical interpretation in terms of principal curvatures and all results derived from it. At an intuitive level, the relation between curvature and vulnerability of explanation methods was already pointed out in their work. We make this relation mathematically precise and derive rigorous bounds from our differential geometrical theory which then enable us to propose effective defense mechanisms.

Figure 5: (a-c) Our method can structurally reproduce a target explanation map, Ghorbani et al cannot. We use the same image as Ghorbani et al for comparability. (d) Similarity to target map (higher is better) and change of winning-class probability for 100 images.

## B  Difference in network output

Figure 6 summarizes the change in the output of the network due to the manipulation. We note that all images have the same classification result as the originals. Furthermore, we note that the change in confidence is small. Last but not least, the norm of the vector of all class probabilities is also very small.

Figure 6: Error analysis of Network output. $\tilde{g}(x)$ denotes pre-activation of the last layer. $g(x)$ is the network output after applying the softmax function to the pre-activation $\tilde{g}(x)$.

## C  Generalization over architectures and data sets

Manipulable explanations are not only a property of the VGG-16 network. In this section, we show that our algorithm to manipulate explanations can also be applied to other architectures and data sets. For the experiments, we optimize the loss function given in the main text. We keep the pre-activation for all network architectures approximately constant, which also leads to approximately constant activation.

### C.1  Additional architectures

In addition to the VGG architecture, we also analyze the explanation's susceptibility to manipulations for the AlexNet, Densenet and ResNet architectures. The hyperparameter choices used in our experiments are summarized in Table 2. We set $\beta_0 = 10$ and $\beta_e = 100$ for beta growth. Only for Densenet we set $\beta_0 = 30$ and $\beta_e = 300$ as for smaller beta values the explanation map produced with softplus does not resemble the explanation map produced with relu. Figure 8 and 7 show that the similarity measures are comparable for all network architectures for the gradient method.

Figure 9, 11, 12 and 10 show one example image for each architecture.

| network | iterations | lr | factors |
|---------|-----------|-----|---------|
| VGG16 | 1500 | $10^{-3}$ | 1e11, 10 |
| AlexNet | 4000 | $10^{-3}$ | 1e11, 10 |
| Densenet-121 | 2000 | $5 \times 10^{-4}$ | 1e11, 10 |
| ResNet-18 | 2000 | $10^{-3}$ | 1e11, 10 |

Table 2: Hyperparameters used in our analysis for all networks.

Figure 7: Change in output for various architectures. $\tilde{g}(x)$ denotes pre-activation of the last layer. $g(x)$ is the network output after applying the softmax function to the pre-activation $\tilde{g}(x)$.

## C.2 Additional datasets

We trained the VGG-16 architecture on the CIFAR-10 dataset[1]. The test accuracy is approximately 92%. We then used our algorithm to manipulate the explanations for the LRP method. The hyperparameters are summarized in Table 3. Two example images can be seen in Figure 13.

Figure 8: Similarity measures for gradient method for various architectures.

| method | iterations | lr | factors |
|--------|-----------|----|---------|
| LRP | 1500 | $2 \times 10^{-4}$ | $10^7, 10^2$ |

Table 3: Hyperparameters used in our analysis for the CIFAR-10 Dataset.

Figure 9: Gradient explanation maps produced with VGG-16 model.

Figure 10: Gradient explanation maps produced with ResNet-18 model.

Figure 11: Gradient explanation maps produced with AlexNet model.

Figure 12: Gradient explanation maps produced with Densenet-121 model.

Figure 13: LRP Method on CIFAR-10 dataset

# D Smoothing explanation methods

One can achieve a smoothing effect when substituting the relu activations for softplus$_\beta$ activations and then applying the usual rules for the different explanation methods.

A smoothing effect can also be achieved by applying the SmoothGrad explanation method, see Figure 14. SmoothGrad adds random perturbations to the image and then averages over the resulting explanation maps. We average over 10 perturbed images with different values for the standard deviation $\sigma$ of the Gaussian noise. The noise level $n$ is related to $\sigma$ by $\sigma = n \cdot (x_{max} - x_{min})$, where $x_{max}$ and $x_{min}$ are the maximum and minimum values the input image can take.

Figure 14: Recovering the original explanation map with SmoothGrad. Left: noise dependence for the correlations of the manipulated explanation (here Gradient and LRP) with the target and original explanation. Line denotes median, $10^{th}$ and $90^{th}$ percentile are shown in semitransparent colour. Center and Right: network input and the respective explanation maps as the noise level is increased for Gradient (center) and LRP (right).

The $\beta$-smoothed or SmoothGrad explanation maps are more robust with respect to manipulations. Figure 15, 16 and 17 show results (MSE, SSIM and PCC) for 100 targeted attacks on the original explanation, the SmoothGrad explanation and the $\beta$-smoothed explanation for explanation methods Gradient and LRP. For all experiments we use the same 100 randomly selected images as previously.

For manipulation of SmoothGrad, we use beta growth with $\beta_0 = 10$ and $\beta_e = 100$. For manipulation of $\beta$-Smoothing, we set $\beta = 0.8$ for all runs. The hyperparameters for SmoothGrad and $\beta$-Smoothing are summarized in Table 4 and Table 5.

| method | iterations | lr | factors |
|--------|-----------|-----|---------|
| Gradient | 1500 | $3 \times 10^{-3}$ | $10^{11}, 10^6$ |
| LRP | 1500 | $3 \times 10^{-4}$ | $10^{11}, 10^6$ |

Table 4: Hyperparameters used in our analysis for SmoothGrad.

| method | iterations | lr | factors |
|--------|-----------|-----|---------|
| Gradient | 500 | $2.5 \times 10^{-4}$ | $10^{11}, 10^6$ |
| Grad x Input | 500 | $2.5 \times 10^{-4}$ | $10^{11}, 10^6$ |
| IntGrad | 200 | $2.5 \times 10^{-3}$ | $10^{11}, 10^6$ |
| LRP | 1500 | $2.0 \times 10^{-4}$ | $10^{11}, 10^6$ |
| GBP | 500 | $5.0 \times 10^{-4}$ | $10^{11}, 10^6$ |
| PA | 500 | $5.0 \times 10^{-4}$ | $10^{11}, 10^6$ |

Table 5: Hyperparameters used in our analysis for $\beta$-smoothing.

In Figure 18 and Figure 19, we directly compare the original explanation methods with the $\beta$-smoothed explanation methods. An increase in robustness can be seen for all methods: explanation maps for $\beta$-smoothed explanations have higher MSE and lower SSIM and PCC than explanation

Figure 15: Left: Similarities between explanations. Markers are mostly right of the diagonal, i.e. the MSE for the smoothed explanations is higher than for the unsmoothed explanations which means the manipulated smoothed explanation map does not closely resemble the target $h^t$. Right: Similarities between images. The MSE for the smoothed methods is greater (right of the diagonal) or comparable (on the diagonal), i.e. greater or comparable perturbations in the manipulated Images when using smoothed explanation methods.

maps for the original methods. The similarity measures for the manipulated images are of comparable magnitude.

## D.1    Pixelflipping

We compare the pixel-flipping performance of the $\beta$-smoothed explanations with SmoothGrad, Gradient and the random baseline. The metric sorts pixels of the validation images according to their importance in the respective explanation and incrementally removes those pixels from the input (here we set the pixels to zero) starting with the most relevant. In each step the networks performance is tested on the complete validation set. A small accuracy then means that the pixels marked relevant by the explanation method were indeed needed for a correct classification, which is a desirable quantity of an explanation method. We set $\beta = 1$ for the $\beta$-smoothed explanations and the noise level for SmoothGrad to $0.2$. Figure 21 shows superior performance of $\beta$-smoothing and SmoothGrad over the original Gradient method.

Figure 16: Left: Similarities between explanations. Markers are mostly left of the diagonal, i.e. the SSIM for the smoothed explanations is lower than for the unsmoothed explanations which means the manipulated smoothed explanation map does not closely resemble the target $h^t$. Right: Similarities between Images. The SSIM for the smoothed methods is lower (left of the diagonal) or comparable (on the diagonal), i.e. bigger or comparable perturbations in the manipulated images when using smoothed explanation methods.

Figure 17: Left: Similarities between explanations. Markers are mostly left of the diagonal, i.e. the PCC for the smoothed explanations is lower than for the unsmoothed explanations which means that the manipulated smoothed explanation map does not closely resemble the target $h^t$. Right: Similarities between Images. The PCC for the smoothed methods is lower (left of the diagonal) or comparable (on the diagonal), i.e. bigger or comparable perturbations in the manipulated images when using smoothed explanation methods.

Figure 18: Comparison of Similarities of Explanation Maps for the original Explanation Methods and the $\beta$-smoothed Explanation Methods. Targeted attacks do not work very well on $\beta$-smoothed explanations, i.e. MSE is higher and SSIM and PCC are lower for the $\beta$-smoothed explanations than for the original explanations.

Figure 19: Comparison of Similarities between original and manipulated images. The similarity measures for images for the $\beta$-smoothed explanation methods are of comparable size or slightly worse (higher MSE, lower SSIM and lower PCC) than for the original explanation method, i.e. the manipulations are more visible for the $\beta$-smoothed explanation methods.

Figure 20: Contour plot of a 2-Layer Neural Network $f(x) = V^\top \mathrm{sp}(W^\top x)$ with $x \in [-1, 1]^2$, $W \in \mathbb{R}^{2 \times 50}$, $V \in \mathbb{R}^{50}$ and $V_i, W_{ij} \sim \mathbb{U}(-1, 1)$. Using a softplus activation with $\beta = 1$ visibly reduces curvature compared to a ReLU activation with $\beta \to \infty$.

Figure 21: Pixelflipping performance compared to random baseline (the lower the accuracy the better the explanation) shows superiority of SmoothGrad and $\beta$-smoothed explanation over the original Gradient method.

# E Proofs

In this section, we collect the proofs of the theorems stated in the main text.

## E.1 Theorem 1

**Theorem 1** *Let* $f : \mathbb{R}^d \to \mathbb{R}$ *be a network with* $sp_\beta$ *non-linearities and* $\mathcal{U}_\epsilon(p) = \{x \in \mathbb{R}^d; \|x - p\| < \epsilon\}$ *an environment of a point* $p \in S$ *such that* $\mathcal{U}_\epsilon(p) \cap S$ *is fully connected. Let* $f$ *have bounded derivatives* $\|\nabla f(x)\| \geq c$ *for all* $x \in \mathcal{U}_\epsilon(p) \cap S$. *It then follows for all* $p_0 \in \mathcal{U}_\epsilon(p) \cap S$ *that*

$$\|h(p) - h(p_0)\| \leq |\lambda_{max}| \, d_g(p, p_0) \leq \beta \, C \, d_g(p, p_0), \tag{3}$$

*where* $\lambda_{max}$ *is the principle curvatures with the largest absolute value for any point in* $\mathcal{U}_\epsilon(p) \cap S$ *and the constant* $C > 0$ *depends on the weights of the neural network.*

**Proof:** This proof will proceed in four steps. We will first bound the Frobenius norm of the Hessian of the network $f$. From this, we will deduce an upper bound on the Frobenius norm of the second fundamental form. This in turn will allow us to bound the largest principle curvature $|\lambda_{max}| = \max\{|\lambda_1| \dots |\lambda_{d-1}|\}$. Finally, we will bound the maximal and minimal change in explanation.

**Step 1:** Let $sp^{(l)}(x) = sp(W^{(l)}x)$ where $W^{(l)}$ are the weights of layer $l$.[2] We note that

$$\partial_k sp(\sum_j W_{ij}x_j) = W_{ik} \, \sigma(\sum_j W_{ij}x_j) \tag{4}$$

$$\partial_l \sigma(\sum_j W_{ij}x_j) = \beta \, W_{il} \, g(\sum_j W_{ij}x_j)) \tag{5}$$

where

$$\sigma(x) = \frac{1}{(1 + e^{-\beta x})}, \qquad\qquad g(x) = \frac{1}{(e^{\beta x/2} + e^{-\beta x/2})^2}. \tag{6}$$

The activation at layer $L$ is then given by

$$a^{(L)}(x) = (sp^{(L)} \circ \cdots \circ sp^{(1)})(x) \tag{7}$$

Its derivative $\partial_k a_i^{(L)}$ is equal to

$$\sum_{s_2 \dots s_L} W_{is_L}^{(L)} \sigma \left( \sum_j W_{ij}^{(L)} a_j^{(L-1)} \right) W_{s_L s_{L-1}}^{(L-1)} \sigma \left( \sum_j W_{s_L j}^{(L-1)} a_j^{(L-2)} \right) \dots W_{s_2 k}^{(1)} \sigma \left( \sum_j W_{s_2 j}^{(1)} x_j \right)$$

We therefore obtain

$$\left\| \nabla a^{(L)} \right\| \leq \prod_{l=1}^{L} \left\| W^{(l)} \right\|_F \tag{8}$$

Deriving the expression for $\partial_k a_i^{(L)}$ again, we obtain

$$\partial_l \partial_k a_i^{(L)} = \sum_m \sum_{s_2 \dots s_L} \{$$

$$W_{is_L}^{(L)} \sigma \left( \sum_j W_{ij}^{(L)} a_j^{(L-1)} \right) W_{s_L s_{L-1}}^{(L-1)} \sigma \left( \sum_j W_{s_L j}^{(L-1)} a_j^{(L-2)} \right)$$

$$\dots \beta \sum_{\hat{s}_m} W_{s_{m+1} \hat{s}_m}^{(m)} W_{s_{m+1} s_m}^{(m)} g \left( \sum_j W_{s_{m+1} j}^{(m)} a_j^{(m-1)}(x) \right) \partial_l a_{\hat{s}_m}^{(m-1)}(x)$$

$$\dots W_{s_2 k}^{(1)} \sigma \left( \sum_j W_{s_2 j}^{(1)} x_j \right) \}$$

We now restrict to the case for which the index $i$ only takes a single value and use $|\sigma(\bullet)| \leq 1$. The Hessian $H_{ij} = \partial_i \partial_j a^L(x)$ is then bounded by

$$\|H\|_F \leq \beta \tilde{C} \tag{9}$$

where the constant is given by

$$\tilde{C} = \sum_m \left\|W^{(L)}\right\|_F \left\|W^{(L-1)}\right\|_F \cdots \left\|W^{(m)}\right\|_F^2 \cdots \left\|W^{(1)}\right\|_F . \tag{10}$$

**Step 2:** Let $e_1 \ldots e_{d-1}$ be a basis of the tangent space $T_p S$. Then the second fundamental form for the hypersurface $f(x) = c$ at point $p$ is given by

$$\mathcal{L}(e_i, e_j) = -\langle D_{e_i} n(p), e_j \rangle \tag{11}$$

$$= -\langle D_{e_i} \frac{\nabla f(p)}{\|\nabla f(p)\|}, e_j \rangle \qquad = -\frac{1}{\|\nabla f(p)\|} \langle H[f] e_i, e_j \rangle + (\ldots) \langle \nabla f(p), e_j \rangle \tag{12}$$

We now use the fact that $\langle \nabla f(p), e_j \rangle = 0$, i.e. the gradient of $f$ is normal to the tangent space. This property was explained in the main text. This allows us to deduce that

$$\mathcal{L}(e_i, e_j) = -\frac{1}{\|\nabla f(p)\|} H[f]_{ij} . \tag{13}$$

**Step 3:** The Frobenius norm of the second fundamental form (considered as a matrix in the sense of step 2) can be written as

$$\|\mathcal{L}\|_F = \sqrt{\lambda_1^2 + \cdots + \lambda_{d-1}^2} , \tag{14}$$

where $\lambda_i$ are the principle curvatures. This property follows from the fact that the second fundamental form is symmetric and can therefore be diagonalized with real eigenvectors, e.g. the principle curvatures. Using the fact that the derivative of the network is bounded from below, $\|\nabla f(p)\| \geq c$, we obtain

$$|\lambda_{max}| \leq \beta \frac{\tilde{C}}{c} . \tag{15}$$

**Step 4:** For $p, p_0 \in \mathcal{U}_\epsilon(p) \cap S$, we choose a curve $\gamma$ with $\gamma(t_0) = p_0$ and $\gamma(t) = p$. Furthermore, we use the notation $u(t) = \dot{\gamma}(t)$. It then follows that

$$n(p) - n(p_0) = \int_{t_0}^t \frac{d}{dt} (n(\gamma(t))) \, dt = \int_{t_0}^t D_{u(t)} n(\gamma(t)) \, dt \tag{16}$$

Using the fact that $D_{u(t)} n(\gamma(t)) \in T_{\gamma(t)} S$ and choosing an orthonormal basis $e_i(t)$ for the tangent spaces, we obtain

$$\int_{t_0}^t D_{u(t)} n(\gamma(t)) \, dt = \int_{t_0}^t \sum_j \langle e_j(t), D_{u(t)} n(\gamma(t)) \rangle \, e_j(t) \, dt = \int_{t_0}^t \sum_j \mathcal{L}(e_j(t), u(t)) \, e_j(t) \, dt \tag{17}$$

The second fundamental form $\mathcal{L}$ is bilinear and therefore

$$\int_{t_0}^t \sum_i \mathcal{L}(e_j(t), u(t)) \, e_j(t) \, dt = \int_{t_0}^t \sum_{i,j} \mathcal{L}(e_j(t), e_i(t)) \, u^i(t) \, e_j(t) \, dt \tag{18}$$

We now use the notation $\mathcal{L}_{ij}(t) = \mathcal{L}(e_j(t), e_i(t))$ and choose its eigenbasis for $e_i(t)$. We then obtain for the difference in the unit normals:

$$n(p) - n(p_0) = \int_{t_0}^t \sum_i \lambda_i(t) \, u^i(t) \, e_i(t) \, dt , \tag{19}$$

where $\lambda_i(t)$ denote the principle curvatures at $\gamma(t)$. By orthonormality of the eigenbasis, it can be easily checked that

$$\langle \sum_i \lambda_i(t) \, u^i(t) \, e_i(t), \sum_j \lambda_j(t) \, u^j(t) \, e_j(t) \rangle \leq |\lambda_{max}|^2 \sum_i u^i(t)^2$$

$$\Rightarrow \left\| \sum_i \lambda_i(t) \, u^i(t) \, e_i(t) \right\| \leq |\lambda_{max}| \, \|u(t)\|$$

Using this relation and the triangle inequality, we then obtain by taking the norm on both sides of (19):

$$\|n(p) - n(p_0)\| \le |\lambda_{max}| \int_{t_0}^{t} \|\dot{\gamma}(t)\| \, dt \,. \tag{20}$$

This inequality holds for any curve connecting $p$ and $p_0$ but the tightest bound follows by choosing $\gamma$ to be the shortest possible path in $\mathcal{U}_\epsilon(p) \cap S$ with length $\int_{t_0}^{t} \|\dot{\gamma}(t)\| \, dt$, i.e. the geodesic distance $d_g(p, p_0)$ on $\mathcal{U}_\epsilon(p) \cap S$. The second inequality of the theorem is obtained by the upper bound on the largest principle curvature $\lambda_{max}$ derived above, i.e. (15).

## E.2 Theorem 2

**Theorem 2** *For one layer neural networks $g(x) = relu(w^T x)$ and $g_\beta(x) = sp_\beta(w^T x)$, it holds that*

$$\mathbb{E}_{\epsilon \sim p_\beta} [\nabla g(x - \epsilon)] = \nabla g_{\frac{\beta}{\|w\|}}(x) \,, \tag{21}$$

*where $p_\beta(\epsilon) = \frac{\beta}{(e^{\beta\epsilon/2} + e^{-\beta\epsilon/2})^2}$.*

**Proof:** We first show that

$$sp_\beta(x) = \mathbb{E}_{\epsilon \sim p_\beta} [relu(x - \epsilon)] \,, \tag{22}$$

for a scalar input $x$. This follows by defining $p(\epsilon)$ implicitly as

$$sp_\beta(x) = \int_{-\infty}^{+\infty} p(\epsilon) \, relu(x - \epsilon) \, d\epsilon \,. \tag{23}$$

Differentiating both sides of this equation with respect to $x$ results in

$$\sigma_\beta(x) = \int_{-\infty}^{+\infty} p(\epsilon) \, \Theta(x - \epsilon) \, d\epsilon = \int_{-\infty}^{x} p(\epsilon) \, d\epsilon \,, \tag{24}$$

where $\Theta(x) = \mathbb{I}(x > 0)$ is the Heaviside step function and $\sigma_\beta(x) = \frac{1}{(1 + e^{-\beta x})}$. Differentiating both sides with respect to $x$ again results in

$$p_\beta(x) = p(x) \,. \tag{25}$$

Therefore, (22) holds. For a vector input $\vec{x}$, we define the distribution of its perturbation $\vec{\epsilon}$ by

$$p_\beta(\vec{\epsilon}) = \prod_i p_\beta(\epsilon_i) \,, \tag{26}$$

where $\epsilon_i$ denotes the components of $\vec{\epsilon}$. We will suppress any arrows denoting vector-valued variables in the following in order to ease notation. We choose an orthogonal basis such that

$$\epsilon = \epsilon_p \hat{w} + \sum_i \epsilon_o^{(i)} \hat{w}_o^{(i)} \qquad \text{with} \qquad \hat{w} \cdot \hat{w}_o^{(i)} = 0 \qquad \text{and} \qquad w = \|w\| \, \hat{w} \,. \tag{27}$$

This allows us to rewrite

$$\mathbb{E}_{\epsilon \sim p_\beta} [relu(w^T(x - \epsilon))] = \mathbb{E}_{\epsilon \sim p_\beta} [relu(w^T x - \|w\| \, \epsilon_p))] = \int p_\beta(\epsilon_p) \left( relu(w^T x - \|w\| \, \epsilon_p) \right) d\epsilon_p$$

By changing the integration variable to $\tilde{\epsilon} = \|w\| \, \epsilon_p$ and using (22), we obtain

$$\mathbb{E}_{\tilde{\epsilon} \sim p_\beta} [relu(w^T x - \tilde{\epsilon})] = sp_{\frac{\beta}{\|w\|}}(w^T x) \,, \tag{28}$$

The theorem then follows by deriving both sides of the equation with respect to $x$.

# F  Additional examples for VGG

Figure 22: Explanation map produced with the Gradient Explanation Method on VGG.

Target Image     Original Image     Manipulated Image     $|x - x_{adv}|$

$h^t$     $h(x)$     $h(x_{adv})$     $|h(x) - h(x_{adv})|$

Target Image     Original Image     Manipulated Image     $|x - x_{adv}|$

$h^t$     $h(x)$     $h(x_{adv})$     $|h(x) - h(x_{adv})|$

Figure 23: Explanation map produced with the Gradient $\times$ Input Explanation Method on VGG.

Figure 24: Explanation map produced with the Integrated Gradients Explanation Method on VGG.

Figure 25: Explanation map produced with the LRP Explanation Method on VGG.

Figure 26: Explanation map produced with the Guided Backpropagation Explanation Method on VGG.

Figure 27: Explanation map produced with the Pattern Attribution Explanation Method on VGG.

## Footnotes

[1]code for training VGG on CIFAR-10 from `https://github.com/chengyangfu/pytorch-vgg-cifar10`

[2]We do not make the dependence of softplus on its $\beta$ parameter explicit to ease notation.