[Reviews · NeurIPS 2019]

Reviewer 1



The main concern with this work is originality; the paper declares introducing the targeted attack method while it was introduced in the previous literature (https://arxiv.org/pdf/1710.10547.pdf, it has also been cited throughout the work) and spends a substantial portion of the paper on the targeted attack method and its results. Although shallowly discussed in the previous literature and very intuitive, the theoretical discussions are very useful and the direction of using differential geometry to analyze the interpretation fragility of ReLu networks sounds promising. I very much enjoyed the theoretical and empirical discussion that relates the SmoothGrad method to the introduced B-Smoothing method. Given the originality concern, I am afraid that the current contributions of the paper, overall, are not enough for a venue like NeurIPS. The discussed material in the paper is also not coherent enough for the paper to be published as it is. Some questions: - How could we argue that by changing the activation of a network for the B-Smoothed maps, the given explanation is still explaining the original network? In other words, we are computing the explanations for a different network? I think comparing the saliency map given by this method to the original saliency in a non-adversarial setting could show that similarity is preserved. (The question is similar to that of accuracy-robustness trade-off of the adversarial defense community) - How does the B-Smoothed explanation method (as a new explanation method) compare to other methods in the metrics related to explanation goodness (such as removing pixels with the order of saliency and tracking the prediction accuracy, ...) - How would training a network with softplus activations from the beginning work? In this case, there is no guarantee that the F-Norm bound of the Hessian of this new network is smaller than that of the same architecture trained with a ReLu activation and it would be hard to argue that the softplus network is more robust? - Just to make sure, in the results section, for the B-Smooethd and Smothgradient results, was the perturbation recreated or the specific explanations or the same perturbation of the ReLU network was used? (Fig 4,5 and 14-17) - How could we use the provided theoretical discussion to compare two trained models' interpretation fragility? _________________________________________ ADDITIONAL COMMENT AFTER AUTHOR RESPONSE: The authors did a very good job at clarifying distinctions from previous work. The given answers are very clear and it is amazing that so many experiments were performed in the short rebuttal period. The score is adjusted to reflect the response.

Reviewer 2



Originality: To the best of my knowledge, adversarial manipulation of explanations was (foreshadowed by previous research but) new. The constrained optimization method introduced to generate adversarial manipulations follows closely the ones used for generating class-adversarial examples, but it is also new. The theoretical analysis is very closely related to ideas of Ghorbani et al. 2017, but it goes quite beyond it. Quality: The paper is of good quality. Its best quality is the intuition that fragility of explanations can be effectively exploited. This is an important message. The methodological contribution is somewhat simple and limited (e.g. the manipulation algorithm only applies to differentiable explainers like input gradients). The theoretical analysis does make up for it. The length of the related work is somewhat surprising, considering the amount of recent work on explainability. Clarity: All ideas are presented clearly. Significance: This paper showcases an important phenomenon; I am confident that it will receive a lot of attention, as it highlights an important phenomenon at the interface between adversarial attacks and explainability.

Reviewer 3



The paper is well written and strikes a balance between theory and practice. I find Fig. 2 very clear but Fig. 1 not so clear as the first figure of the paper: Should add bottom row=explanation maps, and mention adversarial maps, present a proof of concept of bad manipulations in practice, etc. Revise caption. I would prefer a different title like "Adversarial explanations depend on the geometric curvature: Better smoothed than manipulated" In eq. 4, you mean a minus and not a plus for gamma>0? If so, please explain why. line 65: states pointwise multiplication for symbol Can you describe more what is the effects of clamping at each iteration, and why not clamp after a few iterations? line 137: Hyperplane -> Hypersurface Say that 2nd fundamental form is also called embedded curvature and may be state that 1st fundamental form is metric tensor. After line 150, remove second minus (inner product is non-negative) I like fig 19 of SI Talk about manifold foliations in general? Minor edits: \mathrm{adv} Upper letter in bibref 7 and 16

[Author Response · NeurIPS 2019]

We thank the reviewers for their comments.

**Reply to #1:** While we fully acknowledge that the discussion on the relation to Ghorbani et al. has not made this
sufficiently clear, we disagree with the concerns about originality:
We present a method which can manipulate an image to obtain an *arbitrary* target explanation. The methods proposed
by Ghorbani et al, including their "targeted" method, cannot structurally reproduce a heatmap. It can only increase the
accumulated relevance in a certain subsection of the image. As a result, it is not capable to manipulate the heatmap
to closely reproduce a target explanation except for very selected and simple cases. We demonstrate this below.
Fine-grained control of the heatmap is absolutely essential for attacks on explanations. We also note that their attacks
only keep the classification result the same. This leads to significant changes in the network output (see plot d). From
their analysis, it is therefore not clear whether the explanation *or* the network is vulnerable (and the heatmap simply
reflects the relevance of the perturbation faithfully). Our method keeps the output constant which is crucial for the
geometrical interpretation in terms of principal curvatures and all results derived from it, i.e. all of Sec 3+4.
The final ms will contain a careful discussion on the relation to Ghorbani et al. and a substantially streamlined Section
2. Fig. 3 is moved to the SI. Also we will extend Sec 4 by a discussion of the large scale analysis of $\beta$-smoothing,
previously in the SI, and additional pixel flipping results (Samek et al 2017, IEEE) establishing that $\beta$-smoothing
performs better than unsmoothed methods (see plot for a preview).
*Reply to questions:*
• Relevance for relu networks is strongly suggested by relation to SG and indeed confirmed by pixel flipping.
• Beta smoothing increases pixel flipping performance.
• We did not retrain with softplus. We think it is preferable to modify the explanation method since it is less costly.
• In Fig 4, a manipulation of the unsmoothed method is "undone" by smoothing. In all other figures, the smoothed
method is attacked directly.
• If both networks have softplus non-linearities, we can compare their bound (9). Note that its constant C depends on
the weights of the network.
*Summary:* Our paper introduces a novel method allowing total control over the heatmap, it explains this manipulability
in terms of differential geometry and uses these insights to propose an effective defense with theoretical guarantees.
None of these results were contained in Ghorbani et al. We therefore strongly insist on the originality of our paper with
respect to Ghorbani et al.

**Reply to #3:** We completely agree that our algorithm is for differential explainers (remark will be added). The final ms
will contain a detailed comparison to Ghorbani et al (see reply to #1), acknowledging the intuition contained in the
figure. (Sanity Checks for Saliency Maps) is already discussed in the text. Bounding the (local) Lipschitz constant
of the explanation has the disadvantage that it makes the explanation insensitive to *any* small perturbation, e.g. even
those which lead to a substantial change in output. This is clearly undesirable as the heatmap should explain why the
perturbation leads to such a drastic "change of mind" of the network. Our method does not have this problem, since
it only bounds the same output curvature. The final ms will explain the relation to the nice work by (Alvarez-Melis,
Jaakkola) in detail. We fully agree with the indicated relation to PGD. The same 100 images were used for comparability.
We implemented all your suggestions.

**Reply to #4:** Since our adv attacks move on lines of the same output, they are orthogonal to conventional adv attacks
on the classification. They are therefore (locally) independent. We conducted experiments confirming this theoretical
prediction (cosine of angle between perturbations averaged over 100 images is $-1.6{\times}10^{-9}{\pm}1.6{\times}10^{-8}$). We added pixel
flipping results which show that there is indeed a trade-off between robustifying explanations and their performance.
For $\beta \ll 1$, the method is provably more robust but does not perform well. Our choice $\beta \approx 1$ however lies in a sweet
spot leading to better explanations *and* robustness. We implemented all your suggestions.

Figure 1: (a-c) Our method can structurally reproduce a heatmap, Ghorbani et al cannot. We use the same image as
Ghorbani et al for comparability  (d) Similarity to target map (higher is better) and change of winning-class probability
for 100 images.  (e) $\beta$-smoothing leads to superior pixel flipping performance (smaller is better).

[Meta-Review · NeurIPS 2019]

An interesting paper that sparked some good discussion. The reviewers particularly appreciated the case that was made from the rebuttal, the new experiments. The authors are encouraged to include the suggestions from the reviewers in crafting the revised version of the paper.